# The Willingness of Parents to Vaccinate Their Children Aged from Five to under Twelve Years with COVID-19 Vaccines between February and March 2022 in Vietnam

**DOI:** 10.3390/vaccines10111775

**Published:** 2022-10-22

**Authors:** An Hoai Duong, Giang Huong Duong, Hue Thi Pham

**Affiliations:** 1Faculty of Accounting, Finance and Economics, Business School, Griffith University, Brisbane, QLD 4111, Australia; 2Department of Planning and Finance, Thai Nguyen University of Economics and Business Administration, Thai Nguyen 24118, Vietnam; 3Post-graduate Department, Thai Nguyen University of Agriculture and Forestry, Thai Nguyen 24119, Vietnam

**Keywords:** parents’ willingness, COVID-19 pandemic, children aged from 5 to under 12 years, SARS-CoV-2 virus, COVID-19 vaccines

## Abstract

The current study used data surveyed with 5357 parents/guardians (parents would be used to represent both ‘parents’ and ‘guardians’ hereafter) between February and March 2022 in Vietnam to examine their willingness to vaccinate their children with current COVID-19 vaccines. It applied the multinomial logistic regression model to examine the association between the willingness of parents and selected influential factors. In addition, the reasons that made parent hesitant or unwilling to vaccinate their children were investigated. Moreover, it identified parents’ preferences for vaccine origins. Approximately, 75.4% of the parents were willing, 21.3% were hesitant and 3.3% were unwilling to vaccinate their children. The most common reasons that made the parents hesitant or unwilling to vaccinate their children were their concerns about the vaccine safety, efficacy and immunity. The most and the second most preferred vaccines were those developed/originated in the US and EU, respectively. Parents who were more likely to vaccinate their children included those whose children were insured, who regularly vaccinated their children, who belonged to the vaccine priority groups, who possessed sufficient knowledge about the ways to prevent the virus or about the herd immunity, and who perceived that their children might be infected with the virus and whose children were afraid of needles. Parents who were less likely to vaccinate their children included those who were the family main income source, who had savings, and who had tertiary education or higher.

## 1. Introduction

Although COVID-19 vaccine coverage in Vietnam has reached almost 80% of the population [1,2], the vaccines have been administered to the priority groups, adults and adolescents, leaving children aged between 5 and under 12 unvaccinated [3]. These groups are vulnerable as their immunity system is considered weaker than that of adults and adolescents [4,5]. In addition, their daily activities such as school attendance pose a risk of being infected and spreading the virus (especially the Omicron variant and its sub-variant) to others. As of mid-October, 2021, among children aged 5 to 11, there were more than 8300 COVID-19-related hospitalizations and nearly 100 COVID-19 deaths. In fact, COVID-19 ranks as one of the top 10 causes of death for children aged 5 to 11 years old [6]. In February 2022, the number of children who were infected with COVID-19 under the age of 18 recorded in Vietnam accounted for approximately 19.2%—of which the most infected age group was from 6–12 years old, accounting for 8%, from 13–17 years old, accounting for 4.8%, from 0–2 years old accounting for 3.6% and from 3–5 years old accounting for 2.8% [7]. The number of unvaccinated children infected with the SARS-CoV-2 virus increased sharply every day during this period [8]. At the time of the study, the number of children aged from 5 to under 12 years in Vietnam was 11.8 million, which accounted for approximately 12% of the population [9]. The children aged 0–4 in Vietnam accounted for approximately 7.2% [10]. Since this age group was not recommended to vaccinate, to reach the herd immunity to stop the pandemic, children aged from 5 to under 12 years must be vaccinated. Despite the important role of COVID-19 vaccines, many parents and guardians (parents would be used to represent both ‘parents” and ‘guardians’ hereafter) still hesitated or refused to vaccinate their children. Their hesitancy or refusal was understandable as the vaccines were still new. In addition, severe shocks, even deaths and the impact of myocarditis and pericarditis after being vaccinated, have been observed among adults and adolescents and children, respectively [11,12,13]. In addition, misinformation about the long-term impact of the vaccines on the children’ fertility has been spread among parents. In addition, the impact of COVID-19 infection such as anxiety and mental health appeared to inversely influence the willingness to get vaccinated against the SARS-CoV-2 virus [14]. The current study endeavored to identify the level of the willingness of parents to vaccinate their children. It also investigated the determinants of the parents’ willingness. In addition, it investigated the reasons behind the parents’ hesitancy and denial to vaccinate their children. Furthermore, it inspected the parents’ preferences for vaccine origins.

## 2. Research Design, Methodology, and Models

### 2.1. Study Design

Since the dependent variables are categorical, multinomial logistic regressions are applied. The sample size is calculated based on the following formula [15,16]:*n* = 100 + x × i(1)
where:

x is an integer representing the event per variable, and i is the number of independent variables. The recommendation for x is 50. Based on previous studies and the study context, there were 43 factors that can influence the parents’ willingness to vaccinate their children. Therefore, the sample size calculated from Formula (1) was 2250. The sample size was doubled to increase the study power. The final sample was 5357 (accounting for almost 0.05% of children aged from 5 to under 12 years in Vietnam). Stratified random sampling was applied owing to its advantages [17,18]. The criteria used to stratify were six social-economic zones. They were the Northern Midlands and Mountains, Red River Delta, North Central Coast and Central Coast, Central Highlands, South East, and Mekong Delta [19]. The sub-samples were weighted based on the children population aged from 5 to under 12 years in each zone.

Google Forms was used to design the questionnaires. Details of the questionnaires and variable coding are provided in Appendix B. The links to questionnaires were then distributed using email, Facebook, Twitter, Zalo, Viber, and WhatsApp to survey. The surveys were conducted between 20 February and 6 March 2022. Responses were downloaded in Excel format for analysis. The ethical clearance for the current study was approved and issued by the Department of Preventive Medicine, Vietnamese Ministry of Health. No: 150/QĐ-ĐĐNC/YTDP/2022, Protocol/ID: 150/02/22

### 2.2. Research Methodologies, Models, and Variable Description

The parents were asked if the Vietnamese Ministry of Health (MoH), relevant parties, and experts recommended the COVID-19 vaccines and, if the vaccines were free, would they be willing to vaccinate their children. The options/responses included “willing”, “hesitant” and “unwilling”. Since the responses were categorical (0 = unwilling, 1 = hesitant and 2 = willing), a multinomial logistic specification was applied [20] as follows:(2)$$P(Y_{i}=1|X)=\alpha +\overset{43}{\underset{j=1}{\sum }}\beta_{i}X_{i}+\varepsilon_{i}$$
where:

Y_i_ was the willingness of parents to vaccinate their children of the *i*th respondent. Particularly, the zero group, which consisted of respondents who were unwilling to vaccinate their respondents was used as the base group/category, and the other two groups were compared to this group. Therefore, regression results of this group are unavailable.

Previous studies have found that socio-demographic characteristics were the most common factors influencing the willingness to vaccinate. The socio-demographic characteristics in the current study included the relationship of respondents with the children (denotes as X_1_, 1 = mother/father, 0 = guardian), parents’ gender (X_2_, 1 = male), the number of children in the family (X_3_), parents’ age (X_4_, years old), parents’ residency (X_5_, 1 = urban), and parents’ marital status (X_6_, 1 = married/cohabitated), if parents were the family main income source (X_7_, in Vietnamese culture those who earn the main income in the family are normally the ones who make important decisions such as getting children vaccinated, 1 = yes), if parents had savings (X_8_, in Asian culture people are hesitant to talk about their income but more open to tell if they have savings, 1 = yes), parents’ dependents (X_9_, persons, excluding the children), parents’ education (X_10_, 1 = tertiary or above) and parents’ employment (X_11_, 1 = employed with permanent income) [21,22,23,24,25,26,27,28,29,30,31,32,33,34,35,36,37,38,39]. In addition, the willingness of parents whose children were insured and uninsured (X_12_, 1 = insured children) was anticipated to be different [25,28,34,38,40] Similarly, the willingness (of parents) to vaccinate children who went to school (X_13_, 1 = yes) and who did not go to school might not be the same, as those go to school were believed to face a higher risk [38]. The willingness to vaccinate their children between parents who regularly vaccinated (vaccination routine/habit) their children (X_14_, 1 = yes) and those who did not might not be similar [39,41].

The access to information (to receive news on the pandemic and vaccines) was believed to help parents have a sufficient understanding of the pandemic and vaccines, and hence could help them to make the decision. In the current study, the access was represented by the number of information channels that the parents receive news on the pandemic (X_15_, channels). Details of the channels are provided in the Abbreviations and news on the vaccines (X_16_, channels), the frequency to receive news on the pandemic (X_17_, 1 = daily, 0 = otherwise) and news on the vaccines (X_18_, 1 = daily, 0 = otherwise), the sufficiency of information (rated by parents, i.e., perception) on the pandemic (X_19_, 1 = sufficient) and on the vaccines (X_20_, 1 = sufficient), the access to vaccination consultations (X_21_, 1 = yes), and the perception of the consultation sufficiency (X_22_, 1 = yes) [39,41,42].

Parents tend to seek protection for their children if they think their children are vulnerable. In the current study, the children’s vulnerability to the virus was represented by having their parents belonging to the vaccine priority groups (X_23_, 1 = yes) [43] or having someone in the family who belonged to these groups (X_24_, 1 = yes). The children’s vulnerability was also represented by having comorbidities (X_25_, 1 = yes) [44] or having an infected parent (X_33_, 1 = yes) or having an infected family member/friend/parents’ co-worker (X_34_, 1 = yes). The risk of being infected with the virus can also come from school. Therefore, the children’s vulnerability was also represented by having a teacher/instructor/classmate infected with the virus (X_35_, 1 = yes). In addition, the more people in the family who are vaccinated (with COVID-19 vaccines), the less vulnerable the children can be. To examine the impact of this factor, four variables were selected. They were the number of persons in the family who were unvaccinated (X_36_, persons), the number of persons in the family who have had one vaccine shot (X_37_, persons), two vaccine shots (X_38_, persons), and three vaccine shots (X_39_, persons). Similarly, if the children’s teachers/instructors (X_40_, 1 = yes) or friends (X_41_, 1 = yes) were fully vaccinated, the vulnerability (to the virus) of the children could be reduced [21,23,24,29,33,39,42,45].

Knowledge can help parents have a sufficient understanding of the pandemic and virus. Therefore, the willingness to vaccinate the children of parents with sufficient and insufficient knowledge may not be identical [24,25,28,33,42]. In the current study, parents were asked and tested if they had sufficient knowledge of common symptoms of people who were infected with the virus (X_26_, 1 = sufficient), of the virus transmission route (X_27_, 1 = sufficient), of the correct ways to prevent the virus (X_28_, 1 = sufficient), of the spread speed of the virus (X_29_, 1 = sufficient), of the virus fatality (X_30_, 1 = sufficient), of the herd immunity (X_31_, 1 = sufficient), and of the common symptoms after being vaccinated (X_32_, 1 = sufficient).

Fear can also have an impact on the parents’ decisions. Particularly, the willingness to vaccinate the children of parents with fear may not be identical to that of those who are not afraid. In the current study, it included the parents’ fear that their children might catch the virus (X_42_, 1 = afraid, 0 = not afraid) and the children’s fear of needles (X_43_, 1 = afraid, 0 = not afraid). Particularly, parents who were afraid that their children might be infected with the virus were expected to be more likely to vaccinate their children than those without the fear. In contrast, respondents whose children were afraid of needle were expected to be less likely to vaccinate their children [25,26,31,35,39,42]. These factors are briefly illustrated in Figure 1.

## 3. Results

### 3.1. The Willingness of Parents to Vaccinate Their Children

Figure 2 shows that the percentage of parents who were willing to vaccinate their children dominated that of those who were hesitant or unwilling. Yılmaz et al. [39] reported that approximately 36% of the parents were willing to vaccinate their children while almost 64% of them were unwilling to vaccinate their children. Goldman et al. [47] found that approximately 65% of the parents were willing to vaccinate their children while almost 35% of them were unwilling to vaccinate their children. Study sample size, location, and period might contribute to the differences between results in the current study and previous studies.

The willingness of parents with different socio-demographic characteristics to vaccinate their children is presented in Table 1. Table 1 shows two different trends in parents’ willingness to vaccinate their children. The first trend shows that parents’ willingness (to vaccinate their children) in a number of groups was adverse. The groups include parents’ gender, the number of children in the family, parent’s residency, parents’ education, parents’ dependents (excluding the children), parents’ information access to the COVID-19 vaccines, parents’ frequency of receiving news on COVID-19 vaccines, parents’ perception on the sufficiency of the news on COVID-19 pandemic or vaccines, parents’ consultations on COVID-19 vaccines, parents’ perception on the adequacy of the consultations, parents’ exposure to the SARS-CoV-2 virus, the children’s comorbidities, parents’ knowledge sufficiency of the common symptoms of COVID-19 infection or the SARS-CoV-2 virus fatality or herd immunity or the common symptoms after being vaccinated, parents’ close persons SARS-CoV-2 virus infection status, the number of unvaccinated people in the family, children’s friends’ full vaccination, and children’s fear of needles.

The second trend shows that the willingness of parents (to vaccinate their children) in a number of groups was mixed. These groups include age, marital status, economic status (income or savings), employment with income, children’s insurance, children’s school attendance, information access (to COVID-19 pandemic), frequency of receiving news on COVID-19 pandemic, respondent’s knowledge of the SARS-CoV-2 virus transmission or the correct ways to prevent COVID-19 infection, or the SARS-CoV-2 virus spread speed, parents’ SARS-CoV-2 virus infection status, the children’s teachers/instructors’ SARS-CoV-2 virus infection status, the number of people in the family who has had 1, 2, or 3 vaccine shots, children’s teachers and instructors’ full vaccination and parents’ fear of children’s infection with SARS-CoV-2 virus.

### 3.2. Reasons That Made Parents Hesitant or Unwilling to Vaccinate Their Children

The reasons that make parents hesitant or unwilling to vaccinate their children can be used in the vaccination consultations and vaccination communication campaigns. Particularly, the parents were asked why they were hesitant or unwilling to vaccinate their children. Based on the literature and results of the pilots, 15 options were made available for them to select. Results are illustrated in Figure 3.

Figure 3 shows that the three most common reasons that made parents unwilling/hesitant to vaccinate their children were the concern about side effects (Reason 1), the probability that their children might be suffered from myocarditis and pericarditis (Reason 2), and the long-term impact of COVID-19 vaccines on their children’s fertility (Reason 3). Particularly, 76.7% and 50% of the parents who concerned about the side effects of the vaccines were unwilling and hesitant to vaccinate their children, respectively. In addition, 75.8% and 60.2% of the parents were concerned that their children might have suffered from myocarditis and pericarditis were unwilling and hesitant to vaccinate their children, respectively. In addition, 64.8% and 64% of the parents concerned about the long-term impact of COVID-19 vaccines on their children’s fertility were unwilling and hesitant to vaccinate their children, respectively.

The next three most common reasons that discouraged the parents to vaccinate their children included the concern that the current vaccines could not protect their children from new variants/sub-variants (Reason 4), the vaccines were developed for adults (Reason 5), and that vaccinated children could still be infected or re-infected (Reason 6). Particularly, 56.4% and 29.5% of the parents concerned that the current vaccines could not protect their children from new variants/sub-variants were unwilling and hesitant to vaccinate their children, respectively. In addition, 53.4% and 37.2% of the parents concerned that the vaccines were developed for adults were unwilling and hesitant to vaccinate their children, respectively. In addition, 47.8% and 38.6% of the parents who were concerned that vaccinated children could still be infected or re-infected were unwilling and hesitant to vaccinate their children, respectively. The remaining reasons/concerns that made the parents hesitate or refuse to vaccinate their children included that the number of cases and deaths (caused by the virus) in the community was decreasing, insufficient information on the vaccines and the number of cases and deaths (caused by the virus) among children, the perception that the impact of infection on children was less severe than on adults or adolescents, that children had a lower probability of being infected with the virus than adults or adolescents, their children were infected with the virus and did not need to vaccinate, and their children had comorbidities and their children had vaccine reaction history.

### 3.3. Preferences of Parents on Vaccine Origin to Vaccinate Their Children

The parents were asked to state their preferences for vaccine origin to vaccinate their children. Based on the literature and results of the pilots, five options were made available for them to select and they could select multiple options. Results are presented in Table 2.

As shown in Table 2, a majority of the parents (42.6–48.8%) preferred vaccinating their children with vaccines developed in the US. The second most preferred vaccine origin was EU. Vaccines that were developed in Russia, China, and other countries were among the less preferred.

### 3.4. The Association between the Willingness of Parents to Vaccinate Their Children Aged from 5 to under 12 Years and Selected Influential Factors

To mitigate biased regression results, all sources of bias and the ways to deal with bias should be identified and addressed. Correlation among independent variables, or also is known as multicollinearity, can cause the estimated coefficients to be biased. To detect this issue, multicollinearity tests were conducted and the mean of the Variance Inflation Factor (VIF) was 1.07, which is significantly lower than ten (commonly accepted in social science). In addition, the matrix of correlation (details are provided in the Appendix A) showed no correlation coefficients greater than 0.21. These test results indicated that no serious issues of multicollinearity exist [48,49,50,51,52,53,54].

Endogeneity can be another issue that makes the results of the study less or even unreliable. Ideally, methods such as instrumental variables (IV) or Two-stage Least Squares (2SLS) can be used to mitigate the impact of the endogeneity problem. However, an IV is rare in reality [55,56,57]. Similarly, the number of relevant variables that can be used to generate IVs for the 2SLS is not always sufficient [58,59,60]. These tasks appeared to be impossible in the context of the current study. To overcome these challenges, the current study carefully follows previous studies to select the most relevant independent variables.

Although all of the 43 factors/independent variables in the full model can influence the willingness of parents to vaccinate their children, the AIC (Akaike Information Criterion) tests suggest 18 factors/variables that best explain the willingness of parents to vaccinate their children [61,62,63]. Results of the test that give the lowest and second lowest AIC are provided in the Appendix A. The reduced multinomial logistic regression model is as follows:(3)$$P(Y_{i}=1|X)=\alpha +\overset{18}{\underset{j=1}{\sum }}\beta_{i}X_{i}+\varepsilon_{i}$$

*Y_i_* and *X_i_* have been addressed in Equation (2). The second lowest AIC showed no improvement in the coefficients’ significance and odds ratios. The *X_i_* suggested by the AIC tests are presented in the Appendix A. Since the group of parents who were unwilling to vaccinate their children was used as the base group/category (coded as Group zero), regression results of this group are not available. In addition, the willingness of other groups (Groups 1 and 2 represented those who were hesitant and unwilling to vaccinate their children, respectively) was compared with this group. The regression results are presented in Table 3.

Parents who earned the main income in the family were less likely to vaccinate their children. Particularly, the OR, lower, and upper 95% CI of Groups 1 and 2 were 0.589, 0.385–0.900 and 0.688, 0.454–1.043, respectively. The impact was significant at the 5% and 10% levels for Groups 1 and 2, respectively. Similarly, parents who had savings were less likely to vaccinate their children. In particularly, the OR, lower and upper 95% CI of Groups 1 and 2 were 0.569, 0.403–0.803 and 0.676, 0.484–0.945, respectively. The impact was significant at the 1% and 5% level for Groups 1 and 2, respectively. Unexpectedly, parents with tertiary education or above were less likely to vaccinate their children. For example, the OR, lower and upper 95% CI of Groups 1 and 2 were 0.377, 0.216–0.657 and 0.216, 0.152–0.450. The impact was significant at the 1% level for both groups. In contrast, parents whose children were insured were more likely to vaccinate their children. Particularly, the OR, lower and upper 95% CI of Groups 1 and 2 were 3.703, 1.016–13.497 and 3.603, 1.140–11.38. The impact was significant at the 5% level for both groups. Similarly, parents who regularly vaccinated their children were more likely to vaccinate their children. In particular, the OR, lower and upper 95% CI of Groups 1 and 2 were 1.945, 1.136–3.358, and 2.233, 1.323–3.768. The impact was significant at the 5% and 1% levels for Groups 1 and 2, respectively. The vaccination consultations appeared to play an important role in the decision to vaccinate their children. For example, parents who rated the consultation on the COVID-19 vaccination for children ‘sufficient’ were more likely to vaccinate their children (OR = 3.116, 95% CI: 2.216–4.381). The impact was significant at the 1% level for Group 2 but insignificant for Group 1. As anticipated, parents who belonged to the vaccine priority groups were more likely to vaccinate their children. Particularly, the OR, lower and upper 95% CI of Groups 1 and 2 were 1.344, 0.955–1.891 and 1.444, 1.037–2.010, respectively. The impact was at the 10% and 5% levels for Groups 1 and 2, respectively. Parents who sufficiently knew the ways to prevent the virus were more likely to vaccinate their children. In particular, the OR, lower and upper 95% CI of Groups 1 and 2 were 2.289, 1.346–3.892 and 2.517, 1.495–4.237, respectively. The impact was significant at the 1% level for both groups. Likewise, parents who possessed sufficient knowledge of the herd immunity were more likely to vaccinate their children. In particular, the OR, lower, and upper 95% CI of Groups 1 and 2 were 3.213, 1.701–6.069 and 4.991, 2.669–9.336, respectively. The impact was significant at the 1% level for both groups. Parents who were infected with the virus were more likely to vaccinate their children (OR = 1.502, 95% CI: 1.042–2.163). The impact was significant at the 5% level for Group 2 but insignificant for Group 1. Unexpectedly, parents who had a close person (a family member or friend or co-worker) infected with the virus were less likely to vaccinate their children (OR = 0.568, 95% CI: 0.317–1.017). The impact was modest for Group 2 but insignificant for Group 1. As expected, parents who were afraid that their children would be infected with the virus were more likely to vaccinate their children. Particularly, the OR, lower, and upper 95% CI of Groups 1 and 2 were 2.146, 1.716–2.684 and 2.361, 1.908–2.922, respectively. The impact was significant at the 1% level for both groups. Although their children were afraid of needles, the parents were more likely to vaccinate their children. Particularly, the OR, lower, and upper 95% CI of Groups 1 and 2 were 1.337, 1.044–1.713 and 2.169, 1.706–2.757, respectively. The impact was significant at the 5% and 1% levels for Groups 1 and 2, respectively.

As shown in Table 3, male parents were more likely to vaccinate their children (OR = 1.740, 95% CI: 1.090–2.778). The impact was significant at the 5% level for Group 2 but insignificant for Group 1. As expected, older parents were more likely to vaccinate their children (OR = 1.040, 95% CI: 1.010–1.070). The impact was significant at the 1% level for Group 2 but insignificant for Group 1. Unexpectedly, parents who resided in urban areas were less likely to vaccinate their children (OR = 0.485, 95% CI: 0.245–0.960). The impact was significant at the 5% level for Group 2 but insignificant for Group 1.

## 4. Discussion

Figure 2 showed that the percentage of parents who were willing to vaccinate their children aged from 5 to under 12 years dominated that of those who were hesitant and unwilling. This finding indicated that a majority of the parents (that accounted for 75.4%) were aware of the risks that their children might be exposed to and the advantages of the vaccines and vaccination. The most common reasons (illustrated in Figure 3) that made the parents hesitant or unwilling to vaccinate their children included the concern about side effects of the vaccines, the probability that their children might suffer from myocarditis and pericarditis, and the long-term impact of COVID-19 vaccines on their children’s fertility. To encourage these parents to vaccinate their children, their concerns should be taken into account. Side effects such as reaction, severe shocks, or even deaths after being vaccinated have been observed [12,13], and these incidents discouraged the parents from vaccinating their children. Yılmaz et al. [39] reported that almost 77% of the parents refused to vaccinate their children due to the fear of the vaccines side effects while Goldman et al. [47] found that 31% of the parents denied vaccinating their children because of the same reason. The parents were also concerned that their children might suffer from myocarditis and pericarditis symptoms after mRNA COVID-19 vaccination. Although the probability that a child would suffer and the number of children suffering from myocarditis and pericarditis were low, these symptoms have been observed and reported [11]. Although the impact of the vaccines on children’s fertility was incorrect, it still worried parents. Therefore, more studies like that of Wesselink et al. [64] should be conducted, and the results could be used in vaccination consultations and communication campaigns to refute the misinformation. The next three most common reasons that discouraged the parents to vaccinate their children included the concern that the current vaccines could not protect their children from new variants/sub-variants, that they were developed for adults, and that vaccinated children could still be infected or re-infected. The concern about the vaccine efficacy in the current study was in accordance with that discovered by Goldman et al. [65], Leng et al. [66] and Yılmaz et al. [39]. Particularly, Yılmaz et al. [39] and Goldman et al. [65] found that almost 37% and 9% of the parents, respectively, refused to vaccinate their children because of the concern about vaccine efficacy. It has been observed that vaccines could only protect human beings from particular variants, not all variants, especially new variants or sub-variants. In addition, vaccine immunity reduced over time and thus vaccinated people can still be infected. However, the symptoms caused by new or sub-variants in fully vaccinated people were not as severe as those who were unvaccinated [67]. Vaccine communication campaigns could emphasise the advantages of the vaccines.

Although the COVID-19 vaccines used to vaccinate the children were approved and recommended by the MoH, experts and relevant parties, the preferences of the parents on the vaccines varied as shown in Table 2. Particularly, approximately 42.6% to 48.8% of the parents preferred vaccines that were developed in the US to vaccinate their children. This finding was consistent with the authors’ earlier study [68]. The second most preferred (by the parents to vaccinate their children) vaccines were those developed in the EU. The choices of the parents could be explained that the initial immune response to vaccination, the vaccine immunity, and other related issues varied from vaccine to vaccine [69,70,71,72,73,74,75]. This finding implied that using vaccines that were developed in the US or at least in the EU would encourage parents to vaccinate their children.

Table 3 shows that male parents in Group 2 were more likely to vaccinate their children. This finding was in accordance with that discovered by Teasdale et al. [40]. However, Babicki et al. [41] found that the female parents in Poland were more likely to vaccinate their children. This finding implied that additional vaccine and vaccination consultations should target female parents in Vietnam. As the impact of this variable was mixed, it could be further examined in future studies.

As anticipated, older parents in Group 2 who normally possessed considerable knowledge and experience were more likely to vaccinate their children. This finding was almost in agreement with that reported by Babicki et al. [41], Duong et al. [42], Edwards et al. [22], Freeman et al. [23], Kessels et al. [21], and Teasdale et al. [38] but different from that discovered by Alley et al. [76], Al-Mistarehi et al. [29], and Dodd et al. [28]. The finding implied that additional vaccination consultations might be necessary to target younger parents.

Unexpectedly, urban parents in Group 2 were less likely to vaccinate their children. This finding was different from that found by Abedin et al. [27], Duong et al. [42], and Yoda et al. [26]. Urban respondents usually have better information access, hence they might have a better understanding of both the benefits and risks of the vaccines. Perhaps, their concerns about the risks outweighed the perceived benefits. This factor could be further analyzed in future studies.

Parents who earned the main family income were less likely to vaccinate their children. This finding was almost in agreement with that reported by Teasdale et al. [38]. Many parents in this group admitted that they would prefer waiting for vaccines that were developed for children or until evidence of the vaccine side effects and efficacy was sufficiently provided as they had sufficient financial sources to afford that choice. Future studies could further examine the impact of this variable and used income as a proxy for this variable.

Parents who had savings were less likely to vaccinate their children. This finding was opposite what was found by Chew et al. [34], Edwards et al. [22], Khubchandani et al. [31], and Schwarzinger et al. [24]. One possible explanation is similar to that for the main family income.

Parents with tertiary education or higher were less likely to vaccinate their children. This finding was different from that found by Bagateli et al. [37] and Teasdale et al. [38]. Apart from being aware of the risks of the vaccines, these parents might also be aware of the disadvantages or risks of the vaccines and vaccination. Since this finding was unexpected, this variable could be further examined in future studies.

Parents whose children were insured were more likely to vaccinate their children. Dodd et al. [28] found similar vaccination behavior in Australia, but Duong et al. [42] found opposite behavior in ASEAN countries. Similarly, parents who regularly vaccinated their children were more likely to vaccinate their children with COVID-19 vaccines. As previously addressed, if parents have a positive experience with previous vaccination, it can encourage them to vaccinate their children and vice versa [77].

Vaccination consultations proved their necessity. To create more access to the consultation for parents, especially for those who were hesitant or unwilling, contents of the consultations could be made available in multiple formats through multiple channels in multiple languages (currently there are approximately 54 local languages in Vietnam) such as online FAQs (frequently asked questions), printed instructions on leaflets, audio broadcasted via local loudspeakers [78], and videos broadcasted via YouTube.

As expected, parents who belonged to the vaccine priority groups were more likely to vaccinate their children. This finding is almost in agreement with that found by Babicki et al. [41], Duong et al. [42], Freeman et al. [23], Kessels et al. [21], and Kourlaba et al. [33].

COVID-19 vaccines were among the solutions to prevent people from being infected with SARS-CoV-2. Parents who possessed sufficient knowledge of the ways (including vaccination) to prevent COVID-19 infection were more likely to vaccinate their children. This finding indicated that knowledge of methods to prevent the virus was essential and could help parents to make the decision. Therefore, vaccination consultations or communication campaigns should include effective methods to prevent the infection.

Parents who possessed sufficient knowledge of herd immunity were more likely to vaccinate their children. Duong et al. [42], Kourlaba et al. [33], and Schwarzinger et al. [24] found similar behavior. According to WHO, herd immunity can be generated through either vaccination or previous infection [79]. The latter can quickly reduce within a few months after the infection. In addition, the immunity cannot deal with new variants or sub-variants. In addition, Wise [80] found that immunity generated from being infected with Omicron would not be sufficient to prevent humans from future infections. Unless the bivalent COVID-19 vaccines [81] are available and accessible to everyone, such information should be included in vaccine and vaccination consultations and communication campaigns.

As expected, parents (in Group 2) who were infected with the virus were more likely to vaccinate their children. However, parents (also in Group 2) who had a close person (such as a family member or friend or a co-worker) infected with the virus were less likely to vaccinate their children. Having a close person who was infected with the virus poses more risks to children, especially those unvaccinated. Therefore, it was expected to encourage parents to vaccinate their children. This finding differed from that discovered by Duong et al. [42] and Kourlaba et al. [33]. Since this was an unexpected result, it could be further examined in future studies.

Parents who perceived that their children could be infected with the virus were more likely to vaccinate their children. Duong et al. [42], Kelly et al. [25], and Khubchandani et al. [31] examined the association between the participants’ fear of being infected with the virus and the willingness to vaccinate. Duong et al. [42] and Kelly et al. [25] found a significant association, but Khubchandani et al. [31] found an insignificant association.

Parents whose children were afraid of needles were more likely to vaccinate their children. The association was significant at the 1% level. Perhaps, the fear of being infected with the virus dominated that of needles. In addition, more and more friendly vaccination methods have been used such as nasal spray or needle-free [82,83]. Duong et al. [42] found a similar association in ASEAN countries if the pandemic was more severe.

## 5. Conclusions

The current study used data surveyed with 5357 respondents (accounted for approximately 0.05 of the children aged from 5 to under 12 years) between February and March 2022 in Vietnam to examine the willingness of parents to vaccinate their children aged from 5 to under 12 years with COVID-19 vaccines. It applied the multinomial logistic regression model to examine the association between the willingness and selected influential factors.

Approximately, 75.4% of the parents were willing, 21.3% were hesitant, and 3.3% were unwilling to vaccinate their children. This finding indicated that approximately, 24.6% of the parents need additional information about the vaccines and vaccination to make their decision. The most common reasons that made the parents hesitant or unwilling to vaccinate their children included the concern about side effects of the vaccines, the probability that their children might suffer from myocarditis and pericarditis and the long-term impact of COVID-19 vaccines on children’s fertility. The second most common reasons that discouraged the parents to vaccinate their children include concern that the current vaccines could not protect their children from new variants/sub-variants, that the vaccines were developed for adults, and that vaccinated children could still be infected. These findings implied that, to encourage the parents to vaccinate the children, their concern should be comprehensively considered. The most and the second most preferred vaccines were those developed in the US and EU, respectively. This finding implied that using the vaccines they preferred would encourage them to vaccinate their children.

Parents who were more likely to vaccinate their children included those whose children were insured, who regularly vaccinated their children, who belonged to the vaccine priority groups, who possessed sufficient knowledge about the ways to prevent the virus or about the herd immunity, who perceived that their children might be infected with the virus, and whose children were afraid of needles. In addition, the vaccination consultations played an important role in the decision-making of those who were willing to vaccinate their children. Therefore, more access to the consultation would be necessary.

Parents who were less likely to vaccinate their children included those who were the main family income source or had savings or had tertiary education or higher. Due to the limit of the study resources, the study could not cover other countries for which the willingness of parents to vaccinate their children might also be necessary to examine. In addition, the impact of residency and education of parents on their willingness to vaccinate their children was not expected. In addition, gender and age of the children have not been considered in the current study. These factors and factors that may have an inverse impact on the willingness to get vaccinated (as previously addressed) can be further examined in future studies.

## Figures and Tables

**Figure 1 vaccines-10-01775-f001:**
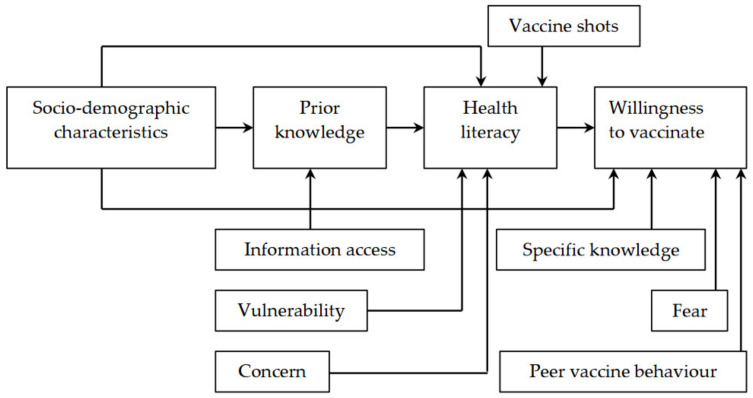
Conceptual framework on parents’ willingness to vaccinate their children aged from 5 to under 12 years. Source. Illustrated by the authors with ideas adapted from Sun et al. [46].

**Figure 2 vaccines-10-01775-f002:**
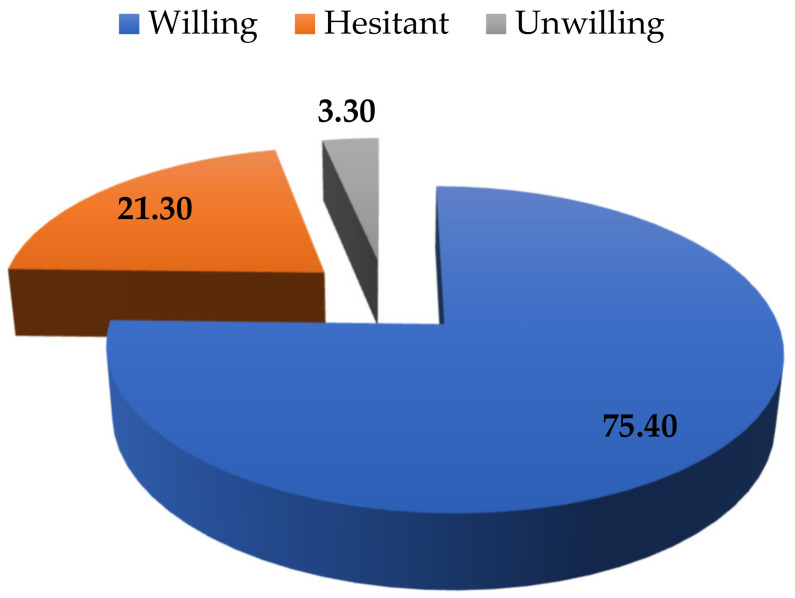
The parents’ willingness to vaccinate their children aged from 5 to under 12 years (measured in percentage). Source. Illustrated by the authors using the surveyed data.

**Figure 3 vaccines-10-01775-f003:**
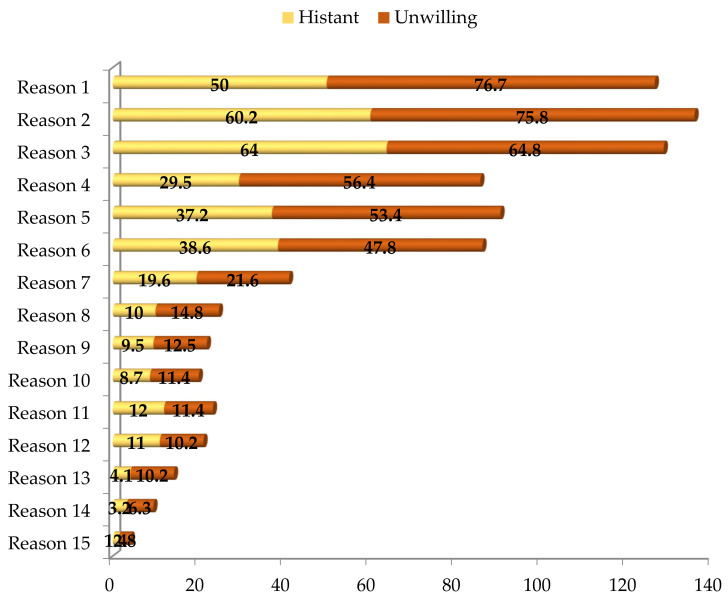
Reasons made parents hesitate or unwilling to vaccinate their children aged from 5 to under 12 years (measured in percentage). Note. Reason 1 = concern about side effects, Reason 2 = concern about the probability that their children might suffer from myocarditis and pericarditis, Reason 3 = concern about the long-term impact of COVID-19 vaccine on children’s fertility, Reason 4 = concern that the current vaccines cannot protect their children from new variants/sub-variants, Reason 5 = the vaccines are developed for adults, Reason 6 = concern that vaccinated children can still be infected, Reason 7 = the number of cases in the community is decreasing, Reason 8 = the number of deaths caused by the virus in the community is decreasing, Reason 9 = there has not been sufficient information on the vaccines, Reason 10 = there has not been sufficient information on the number of cases and deaths caused by the virus among children, Reason 11 = the impact of infection in children is less severe than in adults or adolescents, Reason 12 = children have lower probability of being infected with the virus than adults or adolescents, Reason 13 = my children have been infected with the virus and no need to vaccinate, Reason 14 = my children had comorbidities, Reason 15 = my children had vaccine reaction history. Source. Illustrated by the authors using surveyed data.

**Table 1 vaccines-10-01775-t001:** Willingness of parents to vaccinate their children aged from 5 to under 12 years.

Variable	Measures/ Units	Willing (%)	Hesitant (%)	Unwilling (%)
X_1_^1^	0	4.5	1.8	3.4
	1	95.5	98.2	96.6
X_2_	0	78.2	83.6	86.4
	1	21.8	16.4	13.6
X_3_	≤^2^ 2^4^	83.9	86.6	86.9
	>^3^ 2	16.1	13.4	13.1
X_4_	≤37^5^	57.4	40.7	63.6
	>37	42.6	59.3	36.4
X_5_	0	20.0	10.6	5.7
	1	80.0	89.4	94.3
X_6_	0	8.9	7.6	11.4
	1	91.1	92.4	88.6
X_7_	0	22.4	27.1	18.2
	1	77.6	72.9	81.8
X_8_	0	47.6	52.6	36.9
	1	52.4	47.4	63.1
X_9_	≤2^6^	60.7	57.1	51.1
	>2	39.3	42.9	48.9
X_10_	0	30.3	21.2	9.7
	1	69.7	78.8	90.3
X_11_	0	4.9	5.0	4.0
	1	95.1	95.0	96.0
X_12_	0	1.1	0.7	2.3
	1	98.9	99.3	97.7
X_13_	0	1.0	0.5	0.6
	1	99.0	99.5	99.4
X_14_	0	6.7	7.6	11.9
	1	93.3	92.4	88.1
X_15_	≤6^7^	89.1	85.3	86.9
	>6	10.9	14.7	13.1
X_16_	≤6^8^	91.1	88.3	88.1
	>6	8.9	11.7	11.9
X_17_	0	6.1	10.1	8.0
	1	93.9	89.9	92.0
X_18_	0	17.9	27.6	20.5
	1	82.1	72.4	79.5
X_19_	0	17.5	29.9	21.6
	1	82.5	70.1	78.4
X_20_	0	20.6	38.8	28.4
	1	79.4	61.2	71.6
X_21_	0	41.4	72.2	66.5
	1	58.6	27.8	33.5
X_22_	0	33.2	68.2	64.8
	1	66.8	31.8	35.2
X_23_	0	54.7	56.4	61.4
	1	45.3	43.6	38.6
X_24_	0	63.4	64.6	67.0
	1	36.6	35.4	33.0
X_25_	0	94.0	90.2	90.9
	1	6.0	9.8	9.1
X_26_	0	73.9	67.9	69.9
	1	26.1	32.1	30.1
X_27_	0	58.2	59.4	57.4
	1	41.8	40.6	42.6
X_28_	0	76.4	78.9	90.3
	1	23.6	21.1	9.7
X_29_	0	6.4	6.3	10.2
	1	93.6	93.7	89.8
X_30_	0	6.6	7.2	13.1
	1	93.4	92.8	86.9
X_31_	0	73.7	79.8	93.8
	1	26.3	20.2	6.3
X_32_	0	99.8	99.6	99.4
	1	0.2	0.4	0.6
X_33_	0	71.9	71.3	72.2
	1	28.1	28.7	27.8
X_34_	0	18.7	12.7	8.5
	1	81.3	87.3	91.5
X_35_	0	43.6	38.5	42.0
	1	56.4	61.5	58.0
X_36_	≤2^9^	89.4	89.0	86.4
	>2	10.6	11.0	13.6
X_37_	≤2^10^	69.4	71.5	71.0
	>2	30.6	28.5	29.0
X_38_	≤2^11^	69.0	70.2	68.8
	>2	31.0	29.8	31.3
X_39_	≤3^12^	79.8	79.5	74.4
	>3	20.2	20.5	25.6
X_40_	0	5.1	6.6	5.1
	1	94.9	93.4	94.9
X_41_	0	75.6	79.8	80.1
	1	24.4	20.2	19.9
X_42_	0	6.8	4.5	20.5
	1	93.2	95.5	79.5
X_43_	0	25.9	43.7	58.5
	1	74.1	56.3	41.5

Source. Calculated by the authors using the surveyed data. Note. ^1^ Please refer to the Abbreviations for details, ^2^ Fewer or equal to/younger than, ^3^ More/older than, ^4–12^ The means.

**Table 2 vaccines-10-01775-t002:** Parents’ references for vaccine origins to vaccinate their children aged from 5 to under 12 years.

	Vax 1 ^1^	Vax 2 ^2^	Vax 3 ^3^	Vax 4 ^4^	Vax 5 ^5^
Willing	48.8 ^6^	18.1	8.6	2.0	4.5
Hesitant	44.8	23.0	10.1	1.2	2.8
Unwilling	42.6	25.0	8.5	4.5	2.3

Source. Calculated by the authors using surveyed data. Note. ^1^ Vaccines developed in the US, ^2^ Vaccines developed in EU, ^3^ Vaccines developed in Russia, ^4^ Vaccines developed in China, ^5^ Vaccines developed in other countries and ^6^ this number was calculated by dividing the number of parents who are willing to vaccinate their children with Vax 1 (1971) by the total number of parents who are willing to vaccinate their children (4042).

**Table 3 vaccines-10-01775-t003:** The association between the willingness of parents to vaccinate their children aged from 5 to under 12 years and selected influential factors.

Variable		Sig.^1^	OR ^2^	Effect Size
Group 1	Intercept	0.039	0.019	19.000
	95% Confidence Interval	N/A ^3^	N/A	N/A
	Gender of the respondent (X_2_, 1 = male)	0.117	1.472	0.846
	95% Confidence Interval	0.907 ^4^	2.387 ^5^	0.859
	Age of the respondent (X_4_, years old)	0.384	1.013	0.974
	95% Confidence Interval	0.984	1.044	0.976
	Residency of the respondent (X_5_, 1 = rural)	0.404	0.741	1.528
	95% Confidence Interval	0.367	1.497	1.559
	The respondent was the main family income source (X_7_, 1 = yes)	0.014	0.589	0.856
	95% Confidence Interval	0.385	0.900	0.863
	The respondent had savings (X_8_, 1 = yes)	0.001	0.569	0.842
	95% Confidence Interval	0.403	0.803	0.850
	Education of the respondent (X_10_, 1 = tertiary or above)	0.001	0.377	1.444
	95% Confidence Interval	0.216	0.657	1.460
	The respondents’ children were insured (X_12_, 1 = yes)	0.047	3.703	1.028
	95% Confidence Interval	1.016	13.497	1.185
	Children vaccination habit (X_14_, 1 = yes)	0.015	1.954	0.875
	95% Confidence Interval	1.136	3.358	0.891
	COVID-19 vaccination consultations sufficiency rated by the respondent (X_22_, 1 = sufficient)	0.228	0.804	0.258
	95% Confidence Interval	0.564	1.146	0.262
	The respondent belonged to COVID-19 vaccine priority groups (X_23_, 1 = yes)	0.090	1.344	0.931
	95% Confidence Interval	0.955	1.891	0.941
	The respondent’s knowledge of the ways to prevent SARS-CoV-2 (X_28_, 1 = sufficient)	0.002	2.289	0.909
	95% Confidence Interval	1.346	3.892	0.919
	The respondent’s knowledge of COVID-19 fatality (X_30_, 1 = sufficient)	0.219	1.390	0.928
	95% Confidence Interval	0.822	2.350	0.946
	The respondent’s knowledge of herd immunity (X_31_, 1 = sufficient)	0.000	3.213	0.644
	95% Confidence Interval	1.701	6.069	0.650
	The respondent was infected with SARS-CoV-2 virus (X_33_, 1 = yes)	0.108	1.361	0.906
	95% Confidence Interval	0.935	1.982	0.916
	The respondent had a closed person infected with SARS-CoV-2 virus (X_34_, 1 = yes)	0.128	0.628	1.106
	95% Confidence Interval	0.345	1.143	1.124
	Children’s friends were fully vaccinated (X_41_, 1 = yes)	0.243	1.283	0.920
	95% Confidence Interval	0.845	1.948	0.933
	Respondents were afraid of their children being infected (X_42_, 1 = yes)	0.000	2.146	0.909
	95% Confidence Interval	1.716	2.684	0.919
	Children were afraid of needles (X_43_, 1 = yes)	0.021	1.337	0.616
	95% Confidence Interval	1.044	1.713	0.621
Group 2	Intercept	0.000	0.001	0.053
	95% Confidence Interval	N/A	N/A	N/A
	Gender of the respondent (X_2_, 1 = male)	0.020	1.740	1.182
	95% Confidence Interval	1.090	2.778	1.164
	Age of the respondent (X_4_, years old)	0.008	1.040	1.027
	95% Confidence Interval	1.010	1.070	1.025
	Residency of the respondent (X_5_, 1 = rural)	0.038	0.485	0.655
	95% Confidence Interval	0.245	0.960	0.641
	The respondent was the main family income source (X_7_, 1 = yes)	0.078	0.688	1.168
	95% Confidence Interval	0.454	1.043	1.159
	The respondent had savings (X_8_, 1 = yes)	0.022	0.676	1.188
	95% Confidence Interval	0.484	0.945	1.177
	Education of the respondent (X_10_, 1 = tertiary or above)	0.000	0.261	0.692
	95% Confidence Interval	0.152	0.450	0.685
	Insured children (X_12_, 1 = yes)	0.029	3.603	0.973
	95% Confidence Interval	1.140	11.387	0.844
	Children vaccination habit (X_14_, 1 = yes)	0.003	2.233	1.143
	95% Confidence Interval	1.323	3.768	1.122
	COVID-19 vaccination consultations sufficiency rated by the respondent (X_22_, 1 = sufficient)	0.000	3.116	3.876
	95% Confidence Interval	2.216	4.381	3.823
	The respondent belonged to COVID-19 vaccine priority groups (X_23_, 1 = yes)	0.030	1.444	1.074
	95% Confidence Interval	1.037	2.010	1.063
	The respondent’s knowledge of the ways to prevent SARS-CoV-2 virus (X_28_, 1 = sufficient)	0.001	2.517	1.100
	95% Confidence Interval	1.495	4.237	1.089
	The respondent’s knowledge of COVID-19 fatality (X_30_, 1 = sufficient)	0.117	1.498	1.078
	95% Confidence Interval	0.904	2.483	1.057
	The respondent’s knowledge of herd immunity (X_31_, 1 = sufficient)	0.000	4.991	1.553
	95% Confidence Interval	2.669	9.336	1.538
	The respondent was infected with SARS-CoV-2 virus (X_33_, 1 = yes)	0.029	1.502	1.104
	95% Confidence Interval	1.042	2.163	1.091
	The respondent had a closed person infected with SARS-CoV-2 virus (X_34_, 1 = yes)	0.057	0.568	0.904
	95% Confidence Interval	0.317	1.017	0.890
	Children’s friends were fully vaccinated (X_41_, 1 = yes)	0.106	1.395	1.087
	95% Confidence Interval	0.931	2.088	1.072
	Respondents were afraid of their children being infected (X_42_, 1 = yes)	0.000	2.361	1.100
	95% Confidence Interval	1.908	2.922	1.089
	Children were afraid of needles (X_43_, 1 = yes)	0.000	2.169	1.622
	95% Confidence Interval	1.706	2.757	1.609

Source. Calculated from the surveyed data. Note. ^1^ Significance, ^2^ Odds Ratio and ^3^ Not available or applicable, ^4^ Lower 95% Confidence Interval, ^5^ Upper 95% Confidence Interval, Group 1 = Parents who were hesitant to vaccinate their children and Group 2 = Parents who were unwilling to vaccinate their children.

## Data Availability

The datasets generated and/or analyzed during the current study are not publicly available due to the sensitive nature of the questions asked (especially those on demographic characteristics and personal perceptions and preferences). However, the datasets, test results, and relevant study results generated from SPSS, STATA, and R codes are available from the corresponding author upon a reasonable request.

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
