# Peer review of "The Willingness of Parents to Vaccinate Their Children Aged from Five to under Twelve Years with COVID-19 Vaccines between February and March 2022 in Vietnam"

_vaccines, 2022, doi:10.3390/vaccines10111775_

Round 1

Reviewer 1 Report

Vaccines against SARS-CoV-2 have played the most important role in saving lives in the COVID19 pandemic. Although the SARS-CoV-2 infections are comparatively mild in children but the continuous mutations in SARS-CoV-2 make it more transmissible. Against this background, to prevent the spread of infections, it is important to vaccinate children. The study by Hoai An et al. analyses the interests of parents to vaccinate their children. The obsevations of the study may be useful to revel the public concerns about the vaccines in general. The authors use a multinomial logistic regression model to examine the association between survey parameters. A minor suggestion may increase the reliability of the data.

 Minor suggestion:

The statistical methods used to determine the significance of the results need to be explained in a separate section.

Author Response

Responses or Answers to Reviewer 1 Comments or Questions

Dear Reviewer 1, thank you very much for offering us a great chance to improve our manuscript. We have tried our best to edit it based on your comments and answer your questions as follows:

Point 1: The statistical methods used to determine the significance of the results need to be explained in a separate section.

Response 1: Dear Reviewer 1, thank you very much for your comments. The methods have been explained in Section 2.2. The significance of the results has been discussed in Section 4. We would much be appreciated it and willing to change if you could let us know how to further improve.

Reviewer 2 Report

Duong Hoai An and colleagues surveyed parents' willingness to vaccinate children between the ages of 5 and under 12 years with current COVID-19 vaccines in Vietnam. The results found that about 75.4% of parents were willing. The most common reasons that made parents hesitant or unwilling to vaccinate their children were their concerns about the safety, efficacy, and immunity of the vaccine. The most and second most popular vaccines were those developed/produced in the United States and the European Union, respectively. This survey is interesting, but there are some problems:

1.      What is the reason for the discrepancy between the results of 3.1 and those reported in literatures?

2.      The description of the results in 3.1 and 3.2 is not detailed.

3.      In 3.3, "Reason 3 = Concern about the long-term effects of the New Coronavirus vaccine on children's fertility" is mentioned. Has the gender of children been considered?

4.      Can parents be properly instructed to understand and fill out the survey by using the "link to distribute the survey via email, Facebook, Twitter, Zalo, Viber, and Whatsapp"?

5.      The finding in 3.4 that "parents with a close person (family, friend, or colleague) infected with the virus were less likely to vaccinate their children" is interesting. Was there any subgroup analysis of family, friends, and colleagues?

Author Response

Responses or Answers to Reviewer 2 Comments or Questions

Dear Reviewer 2, thank you very much for offering us a great chance to improve our manuscript. We have tried our best to edit it based on your comments and answer your questions as follows:

Point 1: What is the reason for the discrepancy between the results of 3.1 and those reported in literatures? 

Response 1: Study sample size, location and period may contribute to the differences between results in the current study and previous studies. This has been added to the first paragraph of Section 3.1. All changes are marked with track changes.

Point 2: The description of the results in 3.1 and 3.2 is not detailed 

Response 2: We would much be appreciated it and are willing to change it if you could let us know how to further improve (detail) the description of the results in these two Sections.

Point 3: In 3.3, "Reason 3 = Concern about the long-term effects of the New Coronavirus vaccine on children's fertility" is mentioned. Has the gender of children been considered? 

Response 3: No. The gender of the children has not been considered. Therefore, this is acknowledged as a weakness of the study in the Conclusion with track changes. However, your suggestion will be considered in our future studies.

Point 4: Can parents be properly instructed to understand and fill out the survey by using the "link to distribute the survey via email, Facebook, Twitter, Zalo, Viber, and Whatsapp"? 

Response 4: Yes. As this was an online survey, instructions were very important to assure the quality of the data. The instructions were provided in the Google Forms Description. As addressed in the study, the survey (questionnaires) was designed using Google Forms. Once it was finished, its link was shortened, copied and distributed through such channels. Respondents did not need an email account (except answering using an email address) to answer. In addition, questions that require specific knowledge such as X23, X25-32, and links with sufficient information were inserted in the questions for respondents’ reference.

Point 5: The finding in 3.4 that "parents with a close person (family, friend, or colleague) infected with the virus were less likely to vaccinate their children" is interesting. Was there any subgroup analysis of family, friends, and colleagues? 

Response 5: Our hypothesis was that if the parent/guardian had a close person infected with SARS-CoV-2 virus his/her children would face more risks. However, the impact direction was not as expected. Therefore, future studies can further examine its impact as addressed in the study. It would be ideal to split the sample into subgroups for analyses, but more questions might discourage respondents from answering. Therefore, we have not been able to create subgroups for analysis, but we will keep your idea in mind in future studies.

Reviewer 3 Report

First of all, I am grateful for the opportunity to review this paper. COVID-19 is still an ongoing pandemic that has resulted in global health, economic and social crises. Actually, the vaccination campaign is the first method to counteract the COVID-19 pandemic; however, sufficient vaccination coverage is conditioned by the people’s acceptance of these vaccines. In this context, the first aim of the paper is evaluating the association between the willingness of parents to be vaccinated and the selected influential factors. The second is to examin the reasons for parent hesitant or unwilling to vaccinate their children in Vietnam in 2022.

The subject under study is certainly important, especially in the historical period we are experiencing. The article presents interesting results but, it must be improved, especially for some methodological concerns before being acceptable.

Title: It should be improved, highlighting the main object of the study, period and country.

Introduction: The authors should make it clear about what is the gap in the literature that is filled with this study. The authors must consider other different factors that may be inversely associated to vaccine hesitancy, such as COVID-19 related anxiety and mental health (refer to articles with DOI: 10.3390/ijerph191911929). This factors must be at least cited in the introduction.

Methods: The tool used to acquire information on the studied population a non-standard one. The use of an unreliable instrument is a serious and irreversible limitation. The fact that a similar methods have been used in previous studies is not sufficient. A validation process must be performed and reported to evaluate the tool. What about intelligibility, reliability and validation index?

The authors propose a minimum sample size, but it is not clear to the reader what is the reference population? All Vietnamizes population?

How was questionnaire distributed by the selected social media instruments? It is not clear how participants were recruited? And how did this method allowed to avoid selection bias. Did thy Authors pay some company to provide lists of subscribers or distribute the investigation tool? this still requires detailed explanation.

Statistical analysis: I suggest to insert a measure of the magnitude of the effect for the comparisons. Please consider to include effect sizes.

Ethical Issue: please declare the approval number, that at the moment is reported as XXXX.

Discussion: I suggest expanding the discussion, emphasizing the contribution of the study to the literature. The Authors should add more practical recommendations for the reader, based on their findings. Also, the section of limitations may be improved, the Authors could elaborate on that also by discussing the potential COVID-19 related mental consequence that can affect the vaccine acceptance (refer to above mentioned article).

Author Response

Responses or Answers to Reviewer 3 Comments or Questions

Dear Reviewer 3, thank you very much for offering us a great chance to improve our manuscript. We have tried our best to edit it based on your comments and answer your questions as follows:

Point 1: Title: It should be improved, highlighting the main object of the study, period and country.

Response 1: Thank you for your comments. The study period and location have been added to the Title. All changes are marked with track changes.

Point 2: Introduction: The authors should make it clear about what is the gap in the literature that is filled with this study. The authors must consider other different factors that may be inversely associated to vaccine hesitancy, such as COVID-19 related anxiety and mental health (refer to articles with DOI: 10.3390/ijerph191911929). This factors must be at least cited in the introduction.

Response 2: Dear Reviewer, thank you for your comments. The gaps that the current study tried to fill are stated in the last three sentences of the Introduction. They are:

‘The current study endeavoured to identify the level of the willingness of parents to vaccinate their children. It also investigated the determinants of the parents’ willingness. In addition, it investigated the reasons behind the parents’ hesitancy and denial to vaccinate their children. Furthermore, it inspected the parents’ preferences for vaccine origins.’

Other different factors such as anxiety and mental health, which may have an inverse impact on the willingness to get vaccinated against the SARS-CoV-2 virus, have been addressed in the Introduction. In addition, the recommended study has been cited. Also, the absence of an examination of the inverse impact of these factors has been added to the Conclusion as a weakness of the current study.

Point 3: Methods: The tool used to acquire information on the studied population a non-standard one. The use of an unreliable instrument is a serious and irreversible limitation. The fact that a similar methods have been used in previous studies is not sufficient. A validation process must be performed and reported to evaluate the tool. What about intelligibility, reliability and validation index?

Response 3: We applied the sampling method used by Bujang and colleagues because our regression was logistics and the population included the entire Vietnamese population. Apart from Bujang and colleagues, there are other researchers who used sampling methods that are based on the EPV such as Smeden (https://journals.sagepub.com/doi/full/10.1177/0962280218784726). This study has been cited in the current study. Since our study’s focus was not on sampling, we used the sampling method that best suited our needs.

Point 4: The authors propose a minimum sample size, but it is not clear to the reader what is the reference population? All Vietnamizes population?
How was questionnaire distributed by the selected social media instruments? It is not clear how participants were recruited? And how did this method allowed to avoid selection bias. Did thy Authors pay some company to provide lists of subscribers or distribute the investigation tool? this still requires detailed explanation.

Response 4: Thank you for your questions. According to the sampling method analysed by Bujang and Smeden, the sample size in our study was not minimal. The first paragraph in Section 2.1 shows that the sample was taken from all six economic zones in Vietnam (All Vietnamese population). In addition, the percentage of children aged from 5 to under 12 in each zone was used to stratify. As addressed in the study, the survey (questionnaires) was designed using Google Forms. Once it was finished, its link was shortened, copied and distributed through such channels. Respondents did not need an email account (except answering using an email address) to answer. In addition, questions that require specific knowledge such as X23, X25-32, and links with sufficient information were inserted in the questions for respondents’ references. Ms Pham Thi Hue and Duong Huong Giang with a wide alumni network were in charge of distributing the questionnaires through medical centres in each zone. To minimise the impact of selection bias, the questionnaires were randomly distributed. We did not pay anything to a third party to collect the data.

Point 5: Statistical analysis: I suggest to insert a measure of the magnitude
of the effect for the comparisons. Please consider to include effect sizes

Response 5: Thank you for your suggestion. The effect sizes have been generated and added to Table 3.

Point 6: Ethical Issue: please declare the approval number, that at the
moment is reported as XXXX.

Response 6: Thank you for your comments. The number has been provided in the text where is suitable.

Point 7: Discussion: I suggest expanding the discussion, emphasizing the contribution of the study to the literature. The Authors should add more practical recommendations for the reader, based on their findings. Also, the section of limitations may be improved, the Authors could elaborate on that also by discussing the potential COVID-19 related mental consequence that can affect the vaccine acceptance (refer to above mentioned article).

Response 7: Dear Reviewer, thank you very much for your suggestions. We have tried our best to add as many practical recommendations (which are based on the findings) for the readers as we can. For example, the concerns of parents and vaccine origins/brands. As previously addressed, the absence of an examination of the inverse impact of these factors has been added to the Conclusion as a weakness of the current study.

Round 2

Reviewer 2 Report

It is suggested that Table 1 be properly described in 3.1 of the results.

Author Response

Responses or Answers to Reviewer 2 Comments or Questions

Dear Reviewer 2, thank you very much for offering us another great opportunity to improve our manuscript. We have tried our best to edit it based on your comments as follows:

Point 1: It is suggested that Table 1 be properly described in 3.1 of the results. 

Response 1: Thank you for your comment. Results shown in Table 1 have been briefly interpreted right before and after Table 1 with track changes.

Reviewer 3 Report

The paper was improved and it is now suitable for publication

Author Response

Responses or Answers to Reviewer 3 Comments or Questions

Point 1: The paper was improved and it is now suitable for publication. 

Response 1: Dear Reviewer 3, thank you very much. Your comments have helped us improve our manuscript very much.